# Two Is Company, but Four Is a Party—Challenges of Tetraploidization for Cell Wall Dynamics and Efficient Tip-Growth in Pollen

**DOI:** 10.3390/plants10112382

**Published:** 2021-11-05

**Authors:** Jens Westermann

**Affiliations:** Institute of Molecular Plant Biology, Department of Biology, ETH Zürich, Universitätsstrasse 2, 8092 Zürich, Switzerland; jens.westermann@biol.ethz.ch

**Keywords:** pollen tube, tip-growth, polyploid, genome doubling, cell wall, adaptive evolution, Arabidopsis, signaling, ion homeostasis, secretion, cytoskeleton

## Abstract

Some cells grow by an intricately coordinated process called tip-growth, which allows the formation of long tubular structures by a remarkable increase in cell surface-to-volume ratio and cell expansion across vast distances. On a broad evolutionary scale, tip-growth has been extraordinarily successful, as indicated by its recurrent ‘re-discovery’ throughout evolutionary time in all major land plant taxa which allowed for the functional diversification of tip-growing cell types across gametophytic and sporophytic life-phases. All major land plant lineages have experienced (recurrent) polyploidization events and subsequent re-diploidization that may have positively contributed to plant adaptive evolutionary processes. How individual cells respond to genome-doubling on a shorter evolutionary scale has not been addressed as elaborately. Nevertheless, it is clear that when polyploids first form, they face numerous important challenges that must be overcome for lineages to persist. Evidence in the literature suggests that tip-growth is one of those processes. Here, I discuss the literature to present hypotheses about how polyploidization events may challenge efficient tip-growth and strategies which may overcome them: I first review the complex and multi-layered processes by which tip-growing cells maintain their cell wall integrity and steady growth. I will then discuss how they may be affected by the cellular changes that accompany genome-doubling. Finally, I will depict possible mechanisms polyploid plants may evolve to compensate for the effects caused by genome-doubling to regain diploid-like growth, particularly focusing on cell wall dynamics and the subcellular machinery they are controlled by.

## 1. Tip-Growing Cells as Models to Study Plant Growth Processes

Plants have evolved a fascinating array of cell morphologies and growth behaviors that are essential to their fitness. The manifold forms of growth include bidirectional cell expansion of hypocotyl and root epidermal cells; unidirectional growth of pollen tubes, root hairs, and rhizoids; branching behavior of leaf trichomes and moss protonema filaments; or the puzzle-shaped morphology that is characteristic for dicotyledonous leaf epidermal cells, to name a few. Appropriate cell growth relies on a well-balanced interplay between intracellular turgor pressure and dynamic changes in the biophysical properties of the extracellular cell wall and the intracellular cytoskeleton. Turgor pressure increases with water accumulation in the vacuole, generating a mechanical force acting on the plasma membrane and its cell wall. The cell wall is also referred to as the extracellular matrix. It is mainly composed of polysaccharides (cellulose, pectins, and hemicellulose/xyloglucan) and structural proteins with roles in cell wall rearrangement. The cell needs to balance cell wall rigidity (to avoid loss of cell wall integrity) and flexibility (to allow for turgor pressure-driven growth). This is achieved through constant adjustment of cell wall composition and rearrangement of wall polysaccharides [1,2].

In tip-growing cells, this is a comparatively delicate process, as growth happens unidirectionally through tip-focused cell wall and plasma membrane rearrangement that allows for formation of tubular structures that can cover distances exceeding their width by a thousand-fold and more. The ability to cover vast distances and to significantly increase the cell surface relative to its volume has made tip-growth a process that has been adopted by many cell types across plant lineages and life-phases. Examples are the rooting function of rhizoids and root hairs, photosynthetic activity of moss protonemata, or sperm cell delivery by pollen tubes in seed plants. Pioneer studies on the molecular and cellular processes underlying tip-growth mainly focused on pollen from lily and tobacco, e.g., [3,4,5,6,7,8], followed by research on pollen tubes and root hairs from Arabidopsis, e.g., [9,10,11,12,13,14], but also crop models such as rice, e.g., [15,16]. In the last few years, more comparative studies on related cell types from non-seed plants have become available, extending our view on commonalities and differences of tip-growth control mechanisms across cell types and shedding light on their evolutionary development. This especially includes publications on liverwort rhizoids, e.g., [17,18,19], and moss protonemata, e.g., [20,21,22]. In the scope of this review, I will mainly focus on tip-growth control mechanisms of pollen tubes. Where appropriate (or where information on pollen may be missing), I will supplement my descriptions with insights from other tip-growing cell types.

Pollen is a powerful model system to understand cellular growth processes offering a series of key strengths:Pollen tube growth is comparatively easy to track, as it happens solely at the tip (a feature which is shared with other tip-growing cells including rhizoids and root hairs [23]). Additionally, pollen tube growth dynamics can be studied under in vitro (e.g., used in [24]), in vivo (e.g., used in [25]) and semi-in vivo conditions (e.g., used in [26]), which allows for comparison of a series of pollen traits important for male fertility, such as pollen germination, live-imaging of growth, quantification of ovule targeting, and timely sperm release across different experimental setups.Pollen tube growth happens autarkically; pollen represents the male gametophyte of angiosperms and some gymnosperms. It is composed of the pollen grain harboring the sperm cell(s), forming the germ unit together with the vegetative nucleus of the grain. The autonomous character of pollen allows for easy and fast (i.e., high-throughput) analysis of traits linked to pollen (tube) morphology (grain size, tube length and width, and tube polarity) and growth phenotypes (germination, growth, and bursting rates) in vitro, e.g., [10,15,18,27,28,29,30].Pollen is amenable to immuno- and dye-based labeling (e.g., used in [10,31,32,33]), transgenics (e.g., used in [15,29,34]), availability of intracellular sensors (mainly exploited in *Arabidopsis thaliana*), and setups for efficient live-cell imaging (e.g., [35,36,37]).The central role of pollen in plant reproduction and its implication for plant fertility make it an interesting system for plant breeding and crop improvement. Understanding how cell wall dynamics influence pollen tube growth and cell wall integrity is an essential layer within plant fertility research (e.g., [38,39]).

## 2. The Various Layers of Polar Growth Control in Plants

In angiosperms, the pollen cell wall is amongst the most complex extracellular matrix structures in the plant [40]. During tip-growth, the cell wall needs to be dynamically remodeled to meet the unique requirements of unidirectional cell expansion at the cell tip to guarantee not only targeted growth, but also spatiotemporally well-defined loss of cell wall integrity, facilitating sperm release and faithful delivery of the male gametes to the female ovule [10,27]. Therefore, pollen tube growth control has been intensively studied in the past decades and various regulatory layers have been disclosed, many of which are shared with other tip-growing cell types. Examples include shared reliance on ion dynamics and homeostasis, cytoskeletal rearrangement, and tip-directed vesicular transport of proteins and cell wall materials, as well as dynamics of cell wall composition and changes in its rigidity and flexibility [41,42,43,44,45]. A key question in pollen tube growth control is how these regulatory components are interconnected and translate into a unified cellular growth response. Below, I will review the integrated mechanisms required for proper pollen tube growth (illustrated in Figure 1), before discussing how polyploidy-associated cell size and morphology changes may affect them.

### 2.1. Cytoskeletal Dynamics, Endo- and Exocytosis

The cytoskeleton has long been known to contribute to the astounding variety in plant cell morphology and is a central factor for cell elongation, vesicle transport, and cell wall deposition [46]. In tip-growing cells, the cytoskeletal framework is composed of actin filaments and microtubules, both of which are oriented longitudinally in the tube shank and thus ensure resistance of the cell boundaries against the isobaric turgor pressure, as well as external forces. Towards the tip region, actin filaments and microtubules become increasingly scattered, which is thought to be one factor ensuring turgor-driven unidirectional growth [19,47,48]. Besides its function as a strutting system, the cytoskeleton builds a scaffold along which cargoes can be transported; actin mediates vesicular transport of proteins and cell wall materials to the tube apex to establish and maintain cell polarity [47,48,49]. In the plant cell secretory proteins are transported by selective membrane budding at the endoplasmic reticulum, targeting the Golgi complex, the major hub for protein sorting, where secretory proteins are forwarded to the tip via secretory granules (vesicles) [50]. This transport happens via myosins, motor proteins that tether vesicles to actin filaments and move them along the framework to their destination site [51,52]. In turn, retrograde transport vesicles enable recycling of plasma membrane and cell wall materials and protein sorting [53,54]. The tube apex is defined as the most distal region of the growing pollen tube. It contains the “clear zone”, an inverted 5–15 µm long cone-shaped region in which trafficking is almost exclusively restricted to vesicular transport, thus conferring a smooth appearance in micrographs. In contrast, organelles travel more proximolaterally accounting for a comparatively granular cytoplasm in the respective region [55]. By changing cytoskeletal composition, the cell can control the rate at which vesicular cargo is delivered from and to the tip. Important vesicular cargoes include cell wall components such as pectins and xyloglucans, cellulose synthases, and proteinaceous remodelers such as expansins, extensins, or pectin methylesterases that mediate the integration and rearrangement of wall materials to modulate growth [47,53,56]. In contrast to actin, angiosperm microtubules are not thought to directly contribute to tip-growth progression and have only been discussed in pollen tube short range transport relative to actin [55]. However, they are involved in male germ unit transport towards the tube apex [57]. In angiosperms, callose (a polysaccharide and secondary cell wall component) is deposited in regular distances from the basis of the growing pollen tube and can thus be used as a proxy to determine relative growth rates. The so formed callose plugs define a functional compartment in the most distal part of the tube in which trafficking, organelle, and cell wall integrity, and all further processes vital for growth are maintained, while the proximal regions often collapse. The functional compartment also contains the male germ unit which travels along the longitudinal microtubules in the functional cytoplasm, and which is kept in a similar distance from the tube tip up until controlled sperm release [55,58].

### 2.2. Ion Homeostasis and Signaling

Steady pollen tube tip-growth relies on tip-focused H^+^- and Ca^2+^-gradients, which are thought to be established via tip-localized ion influx through trans-membrane ion channels acting in concert with ion pumps (i.e., Ca^2+^- and H^+^-ATPases) localized in the sub-apical/shank region of the pollen tube [59,60,61,62]. The constant H^+^-flux leads to an acidic tip and an alkaline band in the shank region [59]. This pattern is crucial for differential pectin cross-linking to modulate steady growth (see below). In root hairs, disruption of the H^+^-gradient through tip-alkalinization and hyper-acidification have been linked to growth cessation and loss of cell wall integrity, respectively [63]. In *Nicotiana tabacum* pollen tubes, exclusion of the Autoinhibited Plasma Membrane H^+^-ATPase (NtAHA) from the tube apex defines cell polarity and maintains growth [8]. In *Arabidopsis thaliana*, *At*AHA6/8/9 have recently been demonstrated to regulate pollen tube growth through mediating the cytosolic H^+^-gradient [64,65], while *At*AHA2/7 control root cell expansion and root hair tip-growth [66].

Ca^2+^ exerts a multitude of regulatory functions in- and outside of the growing pollen tube. The extracellular matrix of the pollen tube apex contains a high concentration of pectins, which are secreted as methoxyesters through vesicle transport in an oscillatory pattern that precedes growth [67]. The nature of this oscillatory pattern is still comparatively dubious, and many attempts have been made by researchers to explain how turgor pressure, cell wall remodeling, and third (intermediate) processes may influence oscillatory growth. To date, a leading hypothesis is that, given the isotropic nature of the tube, changes in growth direction and rate may solely arise from changing cellular supportive structures such as the cell wall or the cytoskeleton [68]. In another possible scenario, a (periodic) change in turgor pressure would also be conceivable; while slight changes in turgor pressure during growth have been described to exist, they do not seem to correlate with growth rates [69,70,71], rendering a “constant turgor vs. dynamic cell wall” growth model more likely. One strategy how pollen tube growth rates can be altered is demethylesterification of pectin methoxyesters through the activity of pectin methylesterases (PMEs), whose binding is thought to be fostered by apical acidification [7,59,60,72]. Freshly secreted (non-linked) pectin molecules constantly mix with pectins that have already been integrated in the extracellular matrix via cross-linking, which leads to loosening of the cell wall to allow for turgor-driven growth. Ca^2+^ plays an important role in cross-linking acidic, cell wall-localized pectins to limit cell wall flexibility and prevent loss of cell wall integrity. Glutamate Receptor-like Channels (GLRs) and Cyclic Nucleotide-Gated Channels (CNGCs) 7/8/18 are Ca^2+^-permeable channels that have been implicated in mediating tip-focused Ca^2+^-influx to control pollen germination and tube growth [73,74,75,76,77,78], while CNGC5/6/9/14 regulate root hair polarity and growth [79,80,81]. Autoinhibited Ca^2+^-ATP-ASEs (ACAs) are plasma membrane-localized Ca^2+^-pumps, whose pollen-expressed members ACA7/8/9/10 have specific functions in early pollen development (ACA7) and pollen germination/tube growth (ACA8/9/10) [82,83,84]. Accordingly, ACAs are promising candidates for mediation of subapical Ca^2+^-efflux.

Besides its function in cell wall rearrangement, cytoplasmic Ca^2+^ (Ca^2+^_[cyt.]_) functions as a signaling ion that targets regulatory proteins such as calcium-dependent protein kinases (CPKs), Calmodulin (CaM) and the Respiratory Burst Oxidase (RBOH)-class of NADPH oxidases. It influences various processes, including cytoskeletal dynamics, exocytosis, reactive oxygen species (ROS) production, and ion flux (in an auto- and trans-regulatory manner). The pollen-expressed CPK32 has been demonstrated to bind to and activate CNGC18 to promote growth [85]. Most CNGCs have been demonstrated to physically interact with Ca^2+^-binding CaM [86], a suggested mechanism for their (auto)regulation [87]. In stomata, CPK33 controls stomatal opening by phosphorylating K^+^-channels and modulating K^+^-influx [88]. K^+^ is an important cation in regulation of osmosis and cell turgor and is thus of central importance for steady growth control. Consistently, in pollen, Malectin-like Receptor (MLR)-mediated tip-growth control (discussed in Section 2.3) has been linked to potassium transporter activity [15] and Ca^2+^-availability [25]. MLR-mediated growth control via ERULUS is Ca^2+^-dependent [25]. Production of ROS via pollen-expressed NADPH oxidases RBOHH/J (themselves tip-growth control regulators that act downstream of the MLRs ANXUR1/2 (ANX1/2), is also Ca^2+^-dependent. Loss of function of *rbohh/j* is linked to disruption of Ca^2+^-homeostasis, and elevation of cell wall exocytosis, resulting in irregular growth oscillations, loss of cell wall integrity, and reduced fertility [12,34,89]. Altogether, this suggests a mutual regulatory relationship between ROS- and Ca^2+^_[cyt.]_–mediated signaling processes during pollen tube growth.

### 2.3. Peptide and Hormone Signaling

Various classes of signaling molecules that exert hormone(-like) functions have been demonstrated to affect tip-growth control and cell wall integrity signaling in a male- and/or female-dependent manner. LUREs belong to a subgroup of cysteine-rich polypeptides secreted by the female synergids to mediate spatiotemporal coordination of pollen tube targeting and rupture [90]. Pollen-specific receptor-like kinase 6 (PRK6) senses LURE1, and transmits the information across the cell boundaries to recruit Rho of Plants (ROP) signaling proteins (see below) to the plasma membrane with the goal of facilitating pollen tube targeting to the ovule [91].

Similarly, Rapid Alkalinization Factors (RALFs) are small peptide hormones that modulate pollen tube growth and controlled sperm release. However, different RALF homologs are secreted in a male- and female-specific manner; while male-specific RALF4/19 are thought to maintain steady pollen tube tip-growth to target the ovule, RALF34 is implicated in loss of CWI to deliver the sperm cells upon ovule entry [13,27]. RALF4/19 has been demonstrated to act upstream of MLR-signaling which has gained increasing attention in the past years as it has been closely linked to tip-growth control, e.g., [92,93,94]. The MLR-dependent signaling cascade that controls pollen tube growth encompasses activity of ANX1/2 and their homologs Buddha’s Paper Seal1/2 (BUPS1/2) [10,13,14,27]. ANX-BUPS complex binding to their co-receptors Lorelei-like GPI-anchored Protein2/3 (LLG2/3) is fortified through RALF4 activity to promote pollen tube growth [27,29]. In turn, ANX1/2 activate the ROS-producing NADPH-oxidases Reactive Burst Oxidase Homolog H/J (RBOHH/J) [34] that differentially regulate PTI-like MARIS (MRI) kinase [95] and ATUNIS1/2 (AUN1/2) phosphatase [28] activity to balance growth control at adequate levels. Recent studies in the early diverging land plant *Marchantia polymorpha* have demonstrated that two core components of cell wall integrity signaling, the unique Marchantia MLR and PTI-like homologs *Mp*FER and *Mp*MRI, respectively, hold ancestral functions in rhizoid tip-growth [18,96]. Their structural and functional evolutionary conservation and recurrent co-option as tip-growth control regulators (e.g., in pollen and root hairs; see also [94]) in form of a core signaling module highlights the importance of tip-growth control in plant adaptive evolution.

Another key hormone targeting a multitude of intracellular and apoplastic growth regulators is auxin. While it is crucial for various early pollen developmental processes [97,98,99,100,101], including cell wall pattern formation through the Auxin Response Factor 17 (ARF17) [40], there is additional evidence for its effect on pollen tube growth. External application of auxin has been demonstrated to stimulate pollen tube growth in vitro [102,103]. The pollen-specific, ER-localized auxin efflux carrier PIN8 has been proposed to antagonistically control intracellular auxin homeostasis together with PIN5 [99,104]. Another study suggests that—upon pollination—auxin may foster gene expression of Small Auxin Up RNA62/75, which are required for the translation of transcripts relevant for pollen tube growth [105]. In a global context, auxin promotes pectin polymerization and increases pectin viscosity, while leading to depolymerization of hemicellulosic xyloglucans, altogether fostering cell wall loosening and cell expansion [106]. Furthermore, auxin is often associated with ion flux and signaling, apoplastic ROS accumulation, regulation of osmosis and pH, upregulation of cell wall-related genes such as regulators of pectin-, cellulose-, and hemicellulose/xyloglucan dynamics [107], and promotion of apoplast acidification through activation of H^+^-ATPases via their trans-phosphorylation [108,109]. The promotion of apoplast acidification is commonly associated with activation of cell wall loosening proteins (see above) and promotion of growth. Wall loosening leads to activation of calcium channels and thus Ca^2+^-influx, causing cell wall de-acidification, a counteracting force to auxin-induced acidic growth. Consistently, in *Pyrus pyrifolia*, auxin regulates pollen tube growth via regulation of calcium channels in a pH-/H^+^ ATPase-dependent manner, however demonstrated under artificial auxin conditions [110]. Auxin also activates and promotes the expression of K^+^-channels, which leads to an influx of K^+^ triggering increased water uptake and thus, increased turgor pressure [107]. While many of the above-mentioned auxin-related processes remain to be tested for a comparable auxin-mediated response during pollen tube growth, there is evidence from root hairs to be similarly regulated. Auxin positively regulates gene expression of transcription factors, including Root Hair Defective Six-like 4 (RSL4) and likely RSL2, via several ARFs. RSL4, in turn, upregulates ROS-producing peroxidases and NADPH oxidases of the RBOH family (the latter of which are implicated in tip-growth both in pollen tubes and root hairs), increasing apoplastic ROS levels to control polar growth [9,34,111,112,113,114,115]. Consistently, auxin has been implicated in regulation of ROS accumulation in the apoplast mediated by the MLR Feronia (FER) and ROP signaling during root hair growth and, likely, controlled pollen tube rupture [11,116]. With respect to ROP-signaling, auxin also promotes trafficking of vesicles carrying new cell wall material through RopGEF-mediated rearrangement of the cytoskeleton (i.e., actin filaments and microtubules) [117,118]. It will be exciting to learn through future studies which auxin-mediated processes may have been conserved in tip-growth control of pollen tubes and root hairs.

## 3. Effects of Polyploidization on Tip-Growth and Cell Morphology

Whole genome duplication, resulting in polyploidy, is a recurrent feature of plant evolution. Not only is polyploidy common in many natural species, many important crops such as maize, wheat, potato, cabbage, cotton, banana, strawberry, and coffee are also polyploid. Polyploid plants have increased stress resilience, and larger cell and often organ sizes (including seeds and fruits) and associated biomass, explaining the predominance of polyploid crops in agronomy and agriculture [119,120,121]. From an evolutionary perspective, polyploidization leads to a higher degree of genetic redundancy through genome duplication. While this grants safety against the effects of loss or mutation of essential DNA sequences, it may also allow for genome evolution and neo-/sub-functionalization of duplicated genes to adapt to the challenges posed by changing environments, explaining the recurrence of polyploidy in nature [64,122,123,124,125,126]. Despite this predominance of recent and historical polyploidization events as evident in many angiosperm lineages, how it influences plant performance on a (sub)cellular level (e.g., cell growth and division) is far from being fully understood. However, one consistent consequence of whole genome duplication, which is an immediate effect of the increase in DNA content, is an increase in cell size [121,127,128,129]. Such changes in cell size also lead to alteration of cell geometry and composition, including altered volume-to-surface ratio, that could have diverse and important physiological effects [121]. Changes in cell size associated with ploidy increases likely have important effects on tip-growth. In fact, it is evident from the literature that polyploidy affects pollen grain and tube size, morphology, and performance, the potential associated mechanisms of which I will discuss below.

Doubling of the genome is thought to have two positive main effects on pollen tube performance: Heterosis effect and release of selective pressures: The efficiency at which pollen tubes grow and deliver their sperm cells to the female ovule is termed a haploid performance trait that is vital for plant reproductive fitness. Given the haploid nature of pollen and the competitive character of pollen tube growth during reproduction, the rate at which pollen tubes grow has the potential of evolving quickly in the sense that evolution may promote segregation of alleles that facilitate fast growth. The availability of two allelic copies per gene may lead to release of such selective pressures and protect from the negative effects potentially caused by deleterious alleles [130]. At the same time, it can enable novel allelic interactions and thus increase performance during development and growth. This latter effect would initially only be relevant in allopolyploids, which have a hybrid origin, but could also become relevant in autotetraploids (resulting from within-species genome duplications) over time.Metabolic rate increase: It has been hypothesized that polyploidization may increase metabolic rates through increase in gene expression (e.g., see [131]). In turn, increased metabolic rates are thought to increase pollen tube growth rates and likely positively affect associated traits such as pollen grain size, germination rates, and tube length and width [119,130,132,133,134,135]. There is evidence that ploidy may, in some cases, be correlated with pollen tube growth traits in longer term evolution; comprehensive intra-specific and within-species comparisons in taxa that include polyploid lineages revealed a trend of pollen tube growth rate positively correlating with the age of a polyploid, suggesting there is initial slowing associated with genome duplication that evolution compensates for [130]. The above observations raise the question of why pollen shows a trend of decreased growth in younger polyploids across species and taxa. How exactly does genome duplication present a challenge for efficient pollen tube growth? I hypothesize that perturbation of the intricate tip-growth machinery through alteration of transcriptome size, and cell size and geometry (as direct consequences of genome doubling) represent a hurdle for efficient tip-growth that neo-polyploid plant lineages have to evolutionarily overcome. In order to address this, it is crucial to distinguish the immediate effects of (natural or synthetic) polyploidization (i.e., “neo-polyploidy”) and what may be long-term effects and solutions in evolved lineages (i.e., “evolved/paleo-polyploidy”).

### 3.1. Synthetic and Natural Neo-Polyploids

There is a report of a polyploid maize accession to frequently display pollen developmental defects [136] and newly generated *Linum perenne* tetraploids to display increases in pollen pore/colpi number associated with compromised fertility [137]. *Trifolium pratense* varieties (originating from conventional breeding, but sampled from naturally occurring locations) displayed anomalies in pollen morphology, including germination at multiple colpi, changes in growth direction, tip-focused callose deposition and over-accumulation, and differences in tip vs. shank tube widths [138]. Differential callose deposition may affect maintenance of appropriate turgor pressure levels in relation to cell wall composition and elasticity. Imbalance of both components may impede efficient pollen grain germination and steady, directed pollen tube tip-growth. Strikingly, synthetic (Colchicine-induced) generation of diploid pollen in Populus affects the structure of the cell wall ectexine and accounts for decreased pollen tube growth rates [139]. There is additional evidence for root hair development being affected by genome-doubling as well: in synthetic allotetraploid wheat, changes in the expression levels of orthologs of the Arabidopsis root hair gene RSL2 correlate with differences in root hair length [140]. Interestingly, several (natural and synthetic) autotetraploid *Arabidopsis thaliana* accessions have been described to have a higher root hair density as compared to their diploid relatives [141], pointing at a positive link to tip-growth initiation and suggesting that effects of polyploidy on tip-growth may be cell type-specific.

### 3.2. Evolved/Paleo-Polyploids

In *Arabidopsis arenosa* (an obligate outcrossing relative of *Arabidopsis thaliana* with natural tetraploid lineages sharing a monophyletic origin with their diploid relatives around 30,000 generations ago [142]), naturally evolved tetraploids display diploid-like morphology and growth rates that reach or exceed those of diploid lineages. Contrarily, neo-polyploid pollen displays a series of aberrant phenotypes including changes in grain and tube size, altered tube morphology and impaired cell polarity, altogether affecting their performance during tip-growth [143]. Several independently evolved polyploid Tarasa species display smaller pollen grains than their diploid relatives [144], potentially representing a re-diploidization (and possibly over-compensatory) response to grain size increase in their neo-polyploid ancestors. Effects of polyploidy on morphology and growth of pollen in closely related species of Handroanthus and Betula (the latter of which are frequently hybridizing) include wider tubes with diploid-like wall thickness, but higher wall production rates, which appears to maintain growth rates at diploid-like levels and potentially reflects the consequences of higher metabolic activity upon polyploidization [134]. Further indirect evidence comes from Gossypium (having allopolyploid representatives that diverged around 1 mya [145]); polyploidy has been implicated in increasing the width of cotton fiber (i.e., seed hairs/trichomes) [146], which share certain polar growth characteristics with leaf trichomes and tip-growing cells [147,148].

While it is not clear whether the above-mentioned (short- and long-term) changes in pollen (tube) morphology and tip-growth are directly influenced by differential cell wall dynamics, polyploidy has been demonstrated to affect cell wall composition in different cell types and, on a broader evolutionary scale, cell wall composition of tip-growing cell types has diversified over time [149]. In neo-polyploid *A. thaliana*, ploidy negatively correlates with lignin and cellulose levels in stem tissue, while it positively correlates with content of matrix polysaccharides such as hemicellulose and pectins [150], representing the two most abundant cell wall components of dicotyledons [151]. This is in agreement with a study on Cannabis synthetic polyploids reporting decreased cellulose contents in stem tissue upon tetraploidization [152]. Another study correlates polyploidy with elevated expression of genes related to cell wall structure/composition in Arabidopsis sepals [153]. While pollen walls are specialized towards the unique function of their cells (for instance lacking lignin and being composed of temporally intermediate extensine and intensine layers), a similar increase in hemi-/cellulosic components upon polyploidization is conceivable. Thus, one future approach could include comparison of pollen cell wall profiles between diploid, neo-, and evolved polyploid lineages of a taxum, which would enable us to better understand whether (and if so, how) plants adaptively evolve tip-growth through changing cell wall dynamics or composition. May changes in remodeling of the cell wall by fine-tuning of intracellular growth machinery components be one aspect of functional adaptation of pollen tube tip-growth upon genome doubling?

Altogether, the above-mentioned findings suggest that effects of genome-doubling on pollen tube traits may be manifold, while showing a common trend of affecting components of the tip-growth machinery, be it changes in cell wall composition/dynamics itself, or mechanisms acting with them in concert, such as turgor pressure. A leading hypothesis tries to explain the frequently observed changes in pollen (tube) morphology, tip-growth, and performance by linking the direct positive correlation between nuclear content and cell size increase to elevation of transcriptomic activity and net metabolic rates. In fact, germinating pollen has been described to have ten-fold increased respiratory rates as compared to somatic cell types [154] and thus it has been reasoned that transcription of genes relevant for metabolic rates are likely to be DNA template-limited [131]. Following the logic that the metabolic rates underlying (late) pollen tube growth are maintained at comparably high levels to facilitate the fast growth observed in angiosperm tubes, an increase in gene expression (as a consequence of genome duplication) may perturb the tip-growth machinery in a particularly drastic manner. In order to better address this hypothesis and to fully understand how neo-polyploids respond to challenged pollen tube tip-growth, future studies, including both neo- and evolved polyploid lineages, will be needed. This will enable the comparison of immediate effects of genome duplication (i.e., the evolutionary challenge) vs. the long-term effects (i.e., the adaptive mechanism to overcome the initial challenge). Furthermore, pollen tube growth behavior may be influenced by combinations of different ploidies of the male/female gametophytes, as well as the ploidy of the pistillar tissue, as has been indicated before [155]. Therefore, future studies could include comparisons of the reproductive potential of reciprocal crosses between different genotypes (e.g., diploid vs. (neo)tetraploid) to learn whether potential decreases in reproductive fitness may solely be male-specific phenomena or may rather be attributed to effects of polyploidization on male–female interactions. In the following, I will discuss potential mechanistic strategies neo-polyploids may deploy in order to re-adjust pollen tube tip-growth and address the putatively underlying genetics based on pre-existing whole-genomic data from the polyploid model system *Arabidopsis arenosa*.

## 4. Adaptation of Pollen Development and Growth Control upon Polyploidization

When a plant becomes polyploid, this is thought to lead to a sudden change in cellular processes. Common examples include gene expression, cell metabolism, proliferation, growth, and division. As depicted above, tip-growth may be one of the processes that are drastically affected by polyploidization. Given the direct link between pollen tube performance and plant reproductive efficiency, it is interesting to hypothesize on potential strategies neo-polyploid lineages may use to re-adjust pollen tube tip-growth back to a diploid-like manner. Comparative analyses of whole-genomic sequencing data from naturally evolved diploid and tetraploid *A. arenosa* lineages [156,157,158], sharing a monophyletic origin around 30,000 generations ago [142], have revealed a comprehensive list of genes that show signs of having been under selection in tetraploid *A. arenosa*. A subset of genes (i.e., their *A. thaliana* orthologs) has been functionally linked to tip-growth control and related subcellular processes underlying regulation of growth, cell polarity, and cell wall integrity, which raises the question of their putative importance for adaptive evolution in tetraploid pollen. In the following, I will roughly follow the mechanistic layers described in Section 2 to speculate on the roles of some of these genes in adaptive evolution of tip-growth control in tetraploid pollen (illustrated in Figure 2).

### 4.1. Ion Homeostasis and Signaling

Maintenance of ion homeostasis is essential to maintain steady tip-growth (see Section 2). Perturbation of ion levels (e.g., through changes in their net import or export rates) do not only imbalance rates at which cell wall components are being integrated into the dynamically changing extracellular matrix, but also influence cell osmolarity (and thus turgor pressure), as well as signaling pathways for tip-growth control, altogether affecting growth. Firstly, changes in intracellular ion levels will determine osmolarity, which will differentially modulate cytoplasmic turgor pressure. Several cations, being crucial for this process, are imported via tip-focused influx channels and predominantly shank-localized efflux transporters. Pollen-expressed genes with selection marks that function in ion homeostasis are **AHA8**, **ACA8**, and **KEA5**. 

KEA5 is an ABA-dependent K^+^-antiport exporter with H^+^/K^+^-antiport function that has been implicated in osmotic response. In other cell types, loss of function of KEA homologs has been associated with hypersensitivity to low pH and high K^+^-levels, as well as defective cell wall biosynthesis involving the pectin homogalacturonan [159,160]. Thus, KEA5 likely maintains ion homeostasis to guarantee appropriate cell wall composition [161]. May KEA5 have come under selection to adjust pectin composition of the pollen tube cell wall during growth to compensate for altered neo-polyploid cell physiology, such as differential turgor pressures?

The H^+^-ATPase AHA8 and Ca^2+^-ATPase ACA8 have been implicated before in controlling the tip-focused intracellular H^+^- and Ca^2+^-gradients, respectively (both genes are discussed in detail in Section 2). Seeing them having been under selection suggests processes related to the two respective ion classes also having been case of adaptive evolution. May an AHA8 allelic variant be involved in H^+^-mediated acidification at the pollen tube tip to alter the activity of cell wall regulatory proteins? This might be accompanied by local changes in Ca^2+^-concentration through ACA8, influencing the degree of pectin cross-linking to fine-tune the elasticity/rigidity of the cell wall during tip-growth.

### 4.2. Cytoskeletal Dynamics

The cytoskeleton serves both as a static support against the intra- and extracellular mechanical forces acting on the cell boundaries, and as a scaffold to which vesicles can attach in anterograde and retrograde transport. Vesicle transport is of fundamental importance to adjust both tip-focused secretion of cell wall-/plasma membrane materials and their remodelers, as well as their recycling through endocytosis. A set of pollen-expressed genes with selection marks involved in cytoskeletal dynamics include **CAP1** and **JUL1**.

The Cyclase-Associated Protein1 (CAP1) regulates actin cytoskeletal formation necessary for cell elongation and division [162], likely by functioning as nucleotide exchange factor to recharge actin monomers with ATP and to make them available for recycling in cytoskeleton polymerization [163]. In *Arabidopsis thaliana*, CAP1 is strongly expressed both in root hairs and pollen tubes. An *Atcap1* loss of function allele is associated with reduced germination rate in pollen, reduced pollen tube and root hair growth rates and lengths, as well as actin filament disruption and cytoplasmic disorganization in growing root hairs [164]. Thus, CAP1 likely regulates polar growth through its regulatory function on actin rearrangement and organization. Similarly, JAV1-37 Associated Ubiquitin Ligase 1 (JUL1) localizes to and depolymerizes stomatal microtubules in an ABA-promoted and Ca^2+^/H_2_O_2_-dependent manner [165], which raises the question whether it shows a similar mode of action during pollen cytoskeletal rearrangement. While microtubules do not seem to play a direct role in the continuation of pollen tube growth, they have been implicated in conferring mechanical resistance to the growing cell (and possibly being involved in comparatively short-range transport in the growing tube). Assuming that altered cell geometry (e.g., wider or irregular tubes) would require higher wall production rates (as exemplified in other species [131]), a net increase in actin assembly would be a conceivable solution towards this. Increasing actin assembly could allow higher secretion rates of cell wall components, in turn facilitating increased wall production per time. Likewise, alteration of microtubule dynamics could aid the cell to re-shape its geometry/architecture upon polyploidization. Have genes such as CAP1 and JUL1 been functionally adapted to differentially regulate cytoskeletal arrangement in order to meet the new requirements associated with changes in cell geometry?

### 4.3. Endo- and Exocytosis

The above-mentioned hypothetic mechanisms are in agreement with regulators of vesicle transport showing selection marks in tetraploids as well, including **ALA3/12**, **MYOSIN VIIIB**, and **EPSIN1**.

The two Golgi-localized P-type P4-ATPases/flippases, ALA3 and ALA12, have been implicated in vesicle formation through rearrangement of phospholipids across membrane leaflets [166,167,168]. There is recent evidence for ALA3 to maintain tip-localization of Phosphatidylserine (PS) through Rab-GTP-mediated vesicle targeting to the plasma membrane [169]. With respect to polar growth control, the *irregular trichome branch2* (*itb2*) mutation in ALA3 leads to anomalies in trichome expansion, reduction in primary root growth, increase in root hair length and retardation of pollen germination and tube growth suggesting a general role of ALA3 in polar cell growth [170]. The opposing effects on root hairs and pollen tubes suggest that ALA3 is capable of triggering different responses in the two related types of tip-growing cells. Furthermore, *ala3* loss of function mutant pollen tubes display disorganization of cytoplasmic streaming and a drastic reduction in vesicular speed and progressiveness (50% and 80% of wild-type levels, respectively), which correlates with a delay in germination and the production of shorter tubes, altogether translating into a male-specific fertility decrease and reduction in the seed set [171]. The regulation of polar cell growth through ALA3 has furthermore been linked to PIN/AUXIN-signaling [172]. Another pollen-expressed gene with selection marks, RING/U-box superfamily protein AT5G04460, is a physical interactor of myosin XI and was proposed to have a role in myosin cargo attachment/release [173]. Myosins are motor proteins that play a key role in moving vesicles along the cytoskeletal scaffold. Consistently, myosin XI is required for normal cytoplasmic streaming and polar vesicle transport of auxin and membrane-bound molecules in pollen tubes and root hairs, as well as polar growth processes in other cell types and organs [174]. Have flippase activity through ALA3/12 and myosin regulation been functionally adapted to elevate tip-focused vesicle transport, as it might be enabled through cytoskeletal remodelers such as CAP1 and JUL1?

Interestingly the plant-specific, endoplasmic reticulum-/endosome-localized MYOSIN VIIIB has been implicated in endocytosis and secretion during pollen tube growth [175,176], also coming under selection in tetraploids. Another candidate, EPSIN1, is an accessory protein located in the TGN where it is thought to bind to adaptor protein complex 1 (AP-1) to define a subdomain facilitating membrane bulging and vesicle budding through changes in phosphoinositide composition with the goal of endosomal protein sorting [177]. EPSIN homologs can bind to actin filaments, ubiquitin, and certain cargo molecules [178], rendering their evolutionary adaptation in neo-polyploid pollen tubes likely. Assuming elevated wall production rates and associated metabolic processes, the tip-growing cell would also be required to invest more energy into recycling activities. Did MYOSIN VIIIB and EPSIN1 come under selection to adjust endocytosis and protein sorting to the differential physiological requirements associated with higher cell wall production rates through elevated cytoskeletal dynamics and tip-focused secretion?

### 4.4. Signal Transduction

Functional adjustment of neo-polyploid pollen tube tip-growth may not only require evolving mechanistic compounds themselves, but also their upstream regulators. This might be especially relevant in cases where mechanistic genes are not exclusively pollen-expressed and may exert pleiotropic functions, prohibiting them to evolve freely to meet the differential requirements of tetraploid tip-growth. In these cases, adaptive evolution may target signaling components acting further upstream. Promising genes are manifold and may include **FAB1A**, **REN1**, **AGCK1.5**, **KAPP/RAG1**, **UBP8**, and **ARI8**. 

One common regulatory feature in signal transduction is differential signaling through post-translational modifications [179]. The pollen-expressed, auto-phosphorylatable AGC kinase 1.5 (AGCK1.5) has been demonstrated to control pollen tube growth and polarity via physical interaction with Rho of Plants Guanine Nucleotide Exchange Factor 1 (RopGEF1) [180,181]. RopGEFs negatively regulate the activity of plant-specific Rho-type GTPases, ROPs, and ROP/RopGEF signaling has been closely linked to pollen tube growth and polarity [182,183,184,185,186], one function being the rearrangement of tip-focused F-actin [182]. While, in pollen tubes, a hypothetical link between MLR- and ROP-mediated tip-growth control remains to be elucidated, root hair growth is known to involve auxin-mediated ROP signaling via upstream activity of the Malectin-like receptor kinase Feronia (FER) [11]. Strikingly, root hair-expressed RopGEF3 and pollen-expressed ROP1 Enhancer1 (REN1), both of which regulate ROP signaling activities, are also under selection, with REN1 being important for pollen tube polarity [187]. In the quest for mechanistic roles of AGCK-mediated signaling, it has been shown that AGCK1.5 is functionally redundant to its sister homolog AGCK1.7 in *Arabidopsis thaliana*: *agck1.5 agck1.7* double mutants display drastic changes in tip-focused calcium gradient, actin orientation, and vesicle dynamics in growing pollen tubes, leading to changes in tube morphology and polarity [181]. Differential ROP-signaling through REN1, RopGEFs, and AGCK1.5 might connect all three intracellular features to equip neo-tetraploid pollen with a growth machinery that suits the altered cellular requirements caused by genome doubling.

Ubiquitin-specific Protease 8 (UBP8) and U3 ligase ATARI8 (ARI8) are linked to protein-ubiquitination that is commonly associated with protein sorting and proteasomal degradation. In fact, UBP8 mediates de-ubiquitination of proteins and, while no pollen-related function for UBP8 itself has been reported to date, genetic double-knockout of *ubp3* and *ubp4* causes defects during pollen development (i.e., failure of the second mitotic division and changes in endomembrane and vacuolar architecture), germination in vitro, and strong transmission defects in vivo [188]. Interestingly, UBP14 functions in root hair development [189] suggesting a putative conserved role of UBPs in polar growth processes.

Consistently, the Phosphatidylinositol 3-phosphate 5-kinase (PI3P5K) FAB1A is implicated in regulation of vacuolar acidification and reorganization during pollen development, pollen tube growth and guard cell opening/closure, as well as in roots [190,191,192]. The fact that *fab1a/b* mutant pollen grains collapse and are non-viable supports a function of FAB1A/B in osmotic regulation [190]. Reduction in FAB1A/B expression also impairs vacuolar acidification and endocytosis in Arabidopsis root epidermal cells [191]. While FAB1B/C regulate the rate of ABA-induced stomatal closure, this function remains to be tested for FAB1A [192]. Interestingly, other studies suggest multiple further functions of FAB homologs: FAB1B/D are thought to regulate membrane recycling, vacuolar pH, and homeostatic control of reactive oxygen species (ROS) during pollen tube growth [193], while FAB1A/B have been implicated in recycling of auxin transporters [194] (see below). Taken together, it is conceivable that FAB1A may exert a central role in maintenance of turgor-driven growth through regulation of vacuolar dynamics, while interacting with several key signaling components such as ROS and auxin. May adaptation of ubiquitin- and PI3P5K-mediated signaling lead to changes in vacuolar architecture to help fine-tuning of osmotically driven pollen tube growth in tetraploids? It is conceivable that this could possibly happen in coordination with altered wall composition through KEA5 activity (see above).

Another factor linking signaling to cell wall remodeling activities may be the Kinase Associated Protein Phosphatase/Root Attenuated Growth 1 (KAPP/RAG1). KAPP/RAG1 is a Type-2C protein phosphatase (PP2C) interacting with a subset of receptor-like kinase RLKs, including Cavata1 (CLV1) and Cell Wall Associated Kinase 1 (WAK1) [195] to regulate various processes such as root and shoot meristem formation (CLV1/KAPP) [196] and defense/wounding response by sensing of oligogalacturonides (WAK1/KAPP) [197]. The fact that WAK1 binds to several cell wall polysaccharides, including pectins, in a Ca^2+^-induced manner [198], and a recent similar report from other plant species [199], suggest evolutionary conservation of a WAK1-mediated cell wall sensing/decoding function. It would be exciting to learn whether KAPP/RAG1 interaction with (further) RLKs might play a role in cell wall sensing, possibly of pectins or further materials, and decoding and transduction of environmental (extracellular) signals into the cell interior.

## 5. Conclusions

Here, I summarized the basic processes of tip-growth, and what is known about the effects of cellular changes associated with genome doubling on tip-growth processes. Polyploidization has immediate effects on cell size and geometry, and this seems to also have important effects on tip-growth control. What may be the regulatory cause for altered tip-growth processes and what are the strategies of neo-polyploids to compensate for them? Assuming increased gene expression and metabolic rates to be an initial cause, one would expect to see faster (and possibly more heterogeneous) tube growth and/or wall production, as has been proposed before. Likely, this would include increases in associated mechanistic processes such as vesicle transport, cytoskeletal rearrangement, osmotic dynamics and cell wall remodeling. Future experiments could include assessment of cell wall composition and morphology in neo-polyploids as a proxy of processes that may happen further upstream (i.e., intracellularly).

How may plants cope with such changes? Additionally, in a broad scope, do certain mechanistically relevant cell properties connected to cell wall and cytoskeletal remodeling, vesicle transport, and ion flux commonly come under selection, given their omnipresence in cell growth regulation? May such non-meiotic (and possibly even non-reproductive) changes merely be ‘evolutionary accidents’ (i.e., side-effects of genome doubling and expression changes) that can—once compensated for—grant increased fitness to the evolved tetraploid individual? At this stage, this hypothesis is rather speculative and would require a more global comparison of gene expression patterns of genes with selection marks in combination with phenotypic assessment of associated cell type morphologies, as well as integration of both neo- and evolved polyploids, to distinguish what are immediate effects and associated long-term evolutionary solutions.

## Figures and Tables

**Figure 1 plants-10-02382-f001:**
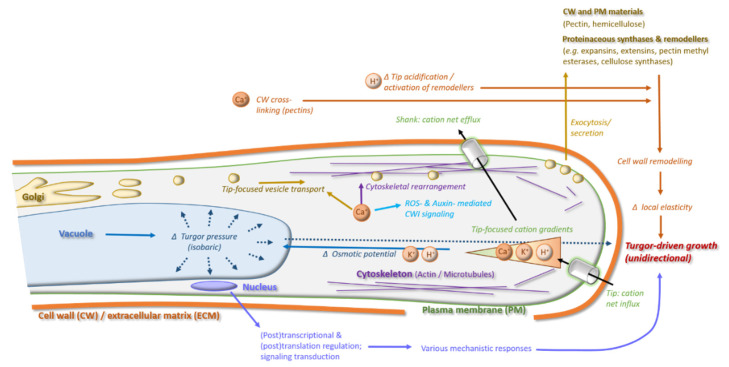
Extra- and intracellular processes and mechanisms underlying modulation of pollen tube cell wall dynamics and tip-growth control.

**Figure 2 plants-10-02382-f002:**
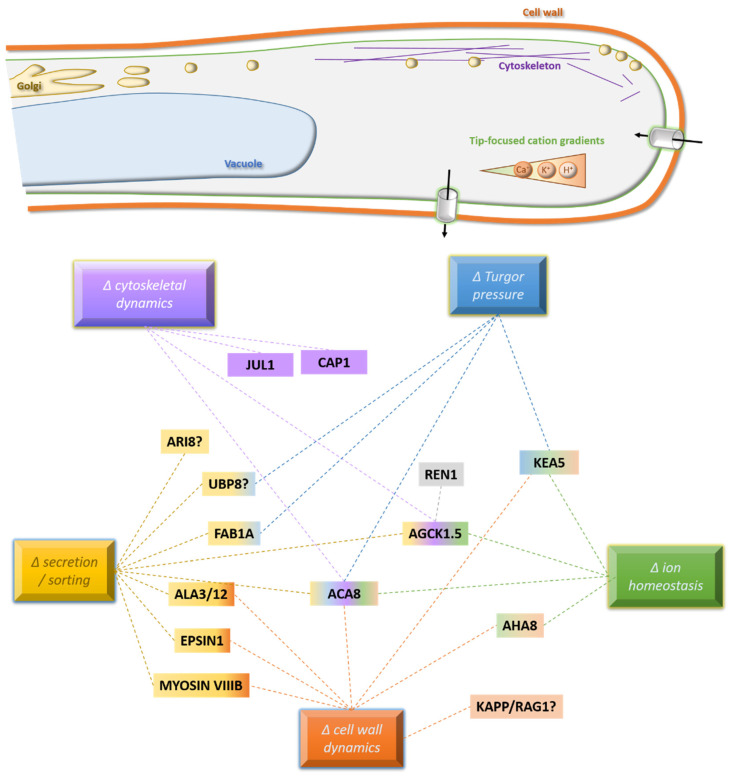
**Hypothetical genetic network underlying the adaptive evolution of tetraploid tip-growth.** The illustration is based on the descriptions in Section 4. The color-code links putative adaptive processes to respective cellular structures involved.

## Data Availability

Data sharing not applicable.

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
