# Peer review of "Two Is Company, but Four Is a Party—Challenges of Tetraploidization for Cell Wall Dynamics and Efficient Tip-Growth in Pollen"

_plants, 2021, doi:10.3390/plants10112382_

Round 1

Reviewer 1 Report

The review of Westermann discusses the phenomenon of plant polyploidization, focusing on the tip growth of pollen tubes. I think it is an exciting topic that was mainly studied from a descriptive/evolutionary point of view and was largely overlooked by cell biologists. After attempting to summarize the main molecular players in tip growth, the author then tries to shed new light on individual genes that were found to be under selective pressure in tetraploid Arabidopsis arenosa lineages and put those into the context of plant cell tip growth. I applaud the author for this effort. However, there are some significant flaws in the manuscript - the information content is quite uneven between chapters, some sections are completely unnecessary (see below), and the general organization of the paper makes it quite difficult to read. For example, the occasional inclusions of the information concerning root hairs or even non-tip growing cells do not add to the central message of the paper (I understand the author’s motive here was to showcase some common features of these tip-growing cells, but I do believe that keeping the manuscript strictly pollen-centric would make better). Generally, the manuscript feels a bit rushed, with some factual errors and omissions (typically, the author frequently mixes the development of pollen grains with the growth of pollen tubes, although these are very different processes). Below I will go section by section:

Specific comments:

1. Tip-growing cells as models to study plant growth processes This is well done, and here the non-pollen data (root hairs, protonemata, rhizoids etc makes sense). I only have one comment - in l. 72 the author states that “Pollen represents the tricellular male gametophyte”, which is obviously not accurate for all species with bicellular pollen, where the final mitosis happens during late pollen tube growth.

2.The various layers of polar growth control in plants In l. 87-89, the author links the complexity of the pollen grain cell wall with the requirements of the tip growth. This is not correct - there is no link between e.g. evolutionary old sporopollenin and the tip growth.

Section 2.1 is described very vaguely, and exocytosis or endocytosis are hardly mentioned except the title. In contrast, section 2.2 is quite detailed with the functional descriptions of individual genes. Since genes from both sections are later discussed in the context of tetraploidization, this lack of balance does not make much sense here.

Somewhat strangely, section 2.3 is the longest, although it is the least relevant for tip growth and/or polyploidy effects. This is especially true for auxin, as its role has been linked to pollen development and not the tip expansion itself. The author cites one study on pear pollen tubes, where auxin was proposed to regulate PT growth via calcium channels to support this putative link. However, this effect was seen only with artificial IAA concentrations and the study itself is not very reliable. Also, the data on RBOHs are somewhat repetitive with section 2.2. I would strongly suggest trimming this section significantly.

3. Effects of polyploidization on tip-growth and cell morphology. The introductory paragraphs and section 3.1 are all very nice - no comments here. On the other hand, section 3.2 is hard to read and some statements are not true: e.g. in l. 375-376, the author writes that “In fact, pollen tubes have been described to have ten-fold increased respiratory rates as compared to somatic cell types ... and thus it has been reasoned that transcription of genes relevant for metabolic rates are likely to be DNA template-limited”. However, the cited studies deal with pollen development, the PT growth! Moreover, at least in angiosperm pollen tubes, the regulation of transcription is in fact a minor strategy compared to the regulation of translation. I urge the author to carefully go over the text to ensure that there are no other misrepresentations in this regard.        

The section 4. Adaptation of pollen development and growth control upon polyploidization is the most important for the whole review as it describes the individual candidate genes undergoing selection in the tetraploids. This is very well done, and I have no comments (except JUL1 - the links to PT growth are too obscure here, as microtubules are not involved in dicot PT tip growth).  

5.Conclusions No comments.

Reviewer 2 Report

The manuscript titled “Two is company, but four is a party – Challenges of tetraploidization for cell wall dynamics and efficient tip-growth in pollen” by Westermann puts forward some interesting hypotheses on pollen tube (PT) growth upon polyploidization. The writing is fine; however, the lack of sufficient illustrative figures (to convey the authors’ view of discussion) makes it poorly readable. Below are some additional suggestions-

  1. As mentioned above, representative figures for at least crucial sub-topics (ion homeostasis, membrane dynamics, signal transduction etc.)- particularly those about polyploid PT growth, would great enhance the readability of the manuscript
  2. The author mentioned to have taken PT as a model tip-growing tissue in plants. While there are crucial similarities between PT and other tip-growing tissues (which the author has mentioned as well), there are fundamental differences between other tip-growing tissues and PT. Most of angiosperm PTs, which traverse long distances, recruit callus plug at interval at its spent growth; the spent growth in physiologically nonfunctional and, in most cases, the spent tube collapses etc. Author is suggested to provide more inclusive view on the subject.
  3. Pollen tube growth is largely oscillatory and the direct role of turgor pressure on it is yet dubious. Ion homeostasis and pH differences addresses it though. Oscillatory growth is typical for other tip-growing tissues as well. The manuscript would benefit from bit more description on the subject.
  4. While the author has discussed on the effect of polyploidization on PT (and pollen) itself, the manuscript lacks the information on PT growth at tissues with different ploidy levels. Some reports indicate that PT growth is retarded in female tissue with higher ploidy level.

Round 2

Reviewer 1 Report

I thank the author for his efforts to make the revised manuscript significantly better. I find it now more balanced, streamlined, and easier to follow. While I’m still not completely sold on the “auxin in pollen tube tip growth” hypothesis, I like the way it is framed now. Thanks for the interesting arguments about the possible transcriptional readjustments in neo-tetraploids in your response - I found it very interesting. While I understand why the author decided not to insert it into this manuscript, it would be interesting to discuss this in more detail.
Overall I’m happy to recommend the revised manuscript for publication.